# Understanding and Visualizing Generalization in UNets

**Abhejit Rajagopal** [1]                                    ABHEJIT.RAJAGOPAL@UCSF.EDU
**Vamshi C. Madala** [2]                                    VAMSHICHOWDARY@UCSB.EDU
**Thomas A. Hope** [1]                                    THOMAS.HOPE@UCSF.EDU
**Peder E. Z. Larson** [1]                                    PEDER.LARSON@UCSF.EDU

[1] *Department of Radiology and Biomedical Imaging, University of California, San Francisco*

[2] *Department of Electrical and Computer Engineering, University of California, Santa Barbara*

## Abstract

Fully-convolutional neural networks, such as 2D and 3D UNets, are now pervasive in medical imaging for semantic segmentation, classification, image denoising, domain translation, and reconstruction. However, evaluation of UNet performance, as with most CNNs, has mostly been relegated to evaluation of a few performance metrics (e.g. accuracy, IoU, SSIM, etc.) using the network's final predictions, which provides little insight into important issues such as generalization and dataset shift that can occur in clinical applications. In this paper, we propose techniques for understanding and visualizing the generalization performance of UNets in image classification and regression tasks, as well as metrics that are indicative of performance on unseen, unlabeled data.

**Keywords:** UNet, generalization, high-dimensional probability, visual representations

## 1. Introduction

Deep learning is well known for its many successes in image recognition and reconstruction, but also for its many pitfalls including overfitting, adversarial examples, and unwanted insertion/deletion of key image content. For medical imaging, in particular, robustness under dataset-shift is a critical issue as algorithms must behave reliably across patient geometries, with evolutions in scanners and scanning protocols, and across clinical imaging sites. The traditional approach to evaluating a model's robustness to such perturbations, or *generalization* performance, is to collect a variety of examples (e.g. from different sites) and evaluate the test accuracy or other relevant metric. The key issue with this approach is that the high dimensionality of the data (e.g. 2D or 3D MRI) makes adequate sampling of different image phenotypes challenging, and evaluation is fundamentally dependent on the availability of annotations. Moreover, it does not address the question of what distinguishes networks that generalize well from those that don't. A satisfying answer to this question would not only improve the interpretability of neural networks, but may induce innovations in architecture and objective function design (Zhang et al., 2016).

In this vein, there is recent interest in developing metrics for predicting the generalization performance of a model directly using test datasets without groundtruth labels, i.e. techniques that *do not depend on measuring the accuracy of the deep learning model*. In particular, recent work in approximation theory indicates that compositional networks with good generalization performance converge in their nodal functions (Poggio et al., 2017; Rajagopal, 2019). This implies that the structure and intermediate computations of DNNs can be used to separate well-performing models from poorly-performing models, independent of where they map on the loss landscape. The key insight of these proofs and empirical

evidence is that, while data may be very sparse in the original input space, the action of a neural network successively projects these to lower dimensional spaces where the data can become dense and where the strong performance guarantees of the theorems can apply.

In this paper, we explore this line of reasoning to develop two new measures of model generalization that do not use any groundtruth data. Specifically, we introduce a new receptive-field analysis of internal feature maps, which not only improves the performance of label-dependent generalization measures in convolutional layers, but enables extension to *arbitrarily large* image sizes encountered in convolutional image-to-image architectures. Crucially for UNets, we derive layer-wise pseudo-labels that provide a geometric interpretation for internal feature maps. This representation enables computationally tractable computation of local class separability, stability, and density from input queries along the interior nodes of a network, which we have found correlate well with test accuracy.

## 1.1. Prior Work

**Uncertainty and Out-of-distribution (OOD) detection.** Popular methods like Monte Carlo dropout (Gal and Ghahramani, 2016), Bootstrap, and ensemble based methods (Hubschneider et al., 2019) are relatively simple, albeit computationally expensive, approaches to estimate the uncertainty and expected generalization of a deep learning model by running multiple inferences, but they are not robust for networks trained with additional regularization. Ren et al. (2019) demonstrated that simple OOD-likelihood estimates using in-distribution data are not reliable. The uncertainty score proposed by Ding et al. (2020) depends only on the final logits of the network, which leaves the model vulnerable to false high-confidence values that can potentially be exploited by the adversarial inputs to the network (Szegedy et al., 2014). Mutual information (MI) using clusters proposed by Peng et al. (2020) rely on a labeled subset of data to impose the consistency across the unlabeled images, a limitation inherent in any MI-based method because the true data distribution is unknown. Negrea et al. (2020) propose improved MI methods but only produce uncertainty bounds for a model that does not extend to the uncertainty of individual samples. This has prompted development of more granular statistical techniques independent of test-data.

**Metrics for Predicting Generalization in Deep Learning (PGDL).** Jiang et al. (2019) evaluated many empirical uncertainty measures such as PAC-Bayes based methods (Dziugaite and Roy, 2017; Neyshabur et al., 2017) and norm-based bounds (Neyshabur et al., 2018), demonstrating the relative success of "sharpness" measures (Keskar et al., 2017) in predicting test accuracy of CNN image classifiers using training data. However, these metrics are prohibitively expensive for large CNNs and image-to-image networks, as they involve computing gradients of the weights with respect to many perturbed inputs.

To this end, the NeurIPS 2020 PGDL Competition (Jiang et al., 2020) was conducted to identify tractable approaches to predict generalization of image classification CNNs. The most successful metrics were based on clustering of the extracted feature space using image-level annotations, and MixUp, a simple method to train the networks on convex combinations of input pairs with the corresponding labels (Zhang et al., 2018). While MixUp performs consistently well in the competition, its applicability to image-to-image networks like UNets is unclear due to degradation of pixel-wise correspondences, especially in quantitative imaging modalities. Further, we note that metrics based on cluster separation and variance are inconsistent in convolutional layers, as shown in Appendix A.1.1 (Figure 5).

### 1.2. Contributions

- We extend interior feature-based analysis of deep neural network classification models to fully-convolutional image-to-image networks, such as 2D and 3D UNets.
- We propose a local receptive field analysis to improve upon existing clustering metrics that are computationally prohibitive and perform poorly with limited data.
- We introduce label-free metrics of generalization ("roughness" and "confidence") that correlate well with test-set performance *without* requiring groundtruth annotations.

## 2. Untangling Deep Feature Representations

We begin with the observation that label-wise clustering metrics, such as those in Natekar and Sharma (2020); Rajagopal et al. (2020), do not generalize well in convolutional layers of simple feed-forward image-based CNNs with scalar target values (Appendix A.1.1). In these approaches, the interior features corresponding to a set of training data are first projected to a lower-dimensional space (e.g. $d = 3$) using principal component analysis (PCA), so each point in Figure 5 represents a single image with corresponding label, indicated by color.

Besides the technical limitations of the particular clustering metrics used, one issue with this cluster-based approach is the size of the training symbols. While the interior features of modern CNNs are indeed *spatially* smaller than the input query (e.g. 32x32 RBG image), they are still quite large and sometimes larger than the input (e.g. 16x16x64 at Layer 1). This presents a computational and performance challenge for feature-based analysis methods, as neither the principle components nor the cluster centers are expected to generalize when the data rate is sufficiently low compared to the dimensionality. To address this, and also enable extensions to UNets that have very high-dimensional feature images, we take a deeper look at the compositional structure and operation of CNNs.

### 2.1. Local Receptive-field Analysis

The real benefit of CNNs is that they *effectively increase the data rate* by using the same interaction kernel over different patches of the image (Agrawal et al., 2019). We use this in-sight to focus the radius of interaction of internal representations to the size of the receptive field. That is, for a network architecture $f^*$, we consider the map from convolutional layer $\ell - 1$ to $\ell$ as the repeated, tiled application of $x_\ell \leftarrow f_\ell(x_{\ell-1})$, where $x_\ell$ denotes the "local" representation of an input image $x$ at layer $\ell$. As an example, for a 3D UNet with single-channel input $x \in \mathbb{R}^{d_1 \times d_2 \times d_3 \times 1}$, we model the map at layer $\ell$ as $f_\ell : \mathbb{R}^{3 \times 3 \times 3 \times k_{\ell-1}} \to \mathbb{R}^{k_\ell}$, where $k_\ell$ represents the number of feature channels at layer $\ell$. Notice that, the input/output dimensionality of $f_\ell$ *does not depend on the size of the input*. Moreover, if we perform PCA at layer $\ell$, we now compress just $k_\ell$ dimensions into $d$ components, whereas applying PCA to the feature images is compressing $\frac{d_1}{2^\ell} \times \frac{d_2}{2^\ell} \times \frac{d_3}{2^\ell}$ dimensions (assuming 2x downsampling in the encoder) which is typically much larger. This alleviates the aforementioned computational bottleneck and results in denser sampling to better represent the node $f_\ell$.

With this compressed representation of the nodes, however, we sacrifice image-level labels. To ameliorate this and maintain the applicability of label-dependent metrics, we create pseudo-labels at each layer by downsampling and quantizing (e.g. `ceiling`) the groundtruth image to the same resolution as the internal feature representation, yielding pixel-wise and patch-wise labels corresponding to the local receptive field. Figure 1 shows the result of this

instrumentation, depicting the geometry of the training data on low-dimensional manifolds at each layer of a UNet trained on a prostate segmentation task from T2-weighted MRI.

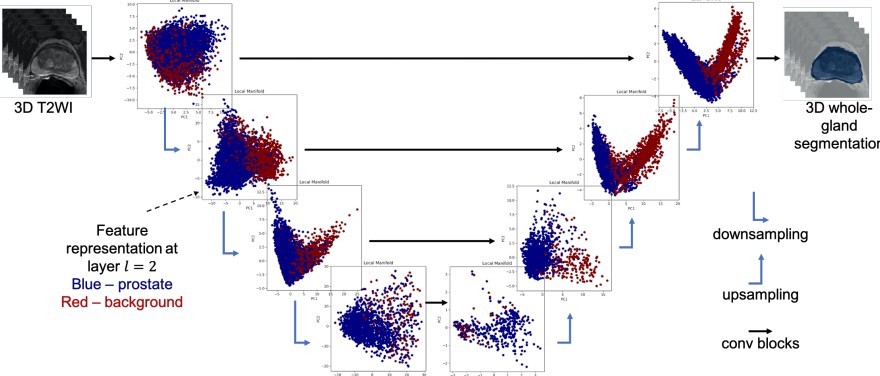

Figure 1: Local receptive-field feature representation of a 3D UNet trained on the binary segmentation of the prostate from T2-weighted MRI, depicting extraction of intermediate feature manifolds with corresponding pseudo-labels indicated by color.

## 2.2. Metrics with Labels

We now outline the best clustering metrics identified in the aforementioned PGDL competition in terms of a UNet's local receptive fields. The same approach can be applied to any feed-forward network with convolutional layers or fully connected layers, although the receptive field analysis described in this paper is tailored for image convolutional layers.

### 2.2.1. Clustering

The central idea of cluster-based methods is that the internal feature space of a DNN can be partitioned to separate semantic classes, and the separability or distortion of this clustering is indicative of a model's generalization performance. The semantic classes can be defined by the label data, as the image classification label or object sub-type, or independently based on other heuristics of the image or feature space. Using the proposed local receptive field analysis, pseudo-labels can be defined on the output space of each convolutional or fully-connected layer $\ell$, enabling a *pixel-wise* clustering of the pixels in $\ell$ or patches in $\ell-1$.

### 2.2.2. Davies-Bouldin (DB) Index Score

The Davies-Bouldin (DB) index score measures the ratio of a clustering's within-cluster distances to between-cluster distances (Davies and Bouldin, 1979), where lower scores indicate more separated clusters, and thus better clustering. To compute DB at an interior layer, we must first choose a clustering of the samples. If the samples are chosen using the aforementioned receptive-field analysis, the clustering can be chosen using the value of the pseudo-label in the output space of each layer. That is, at layer $\ell$, with input feature representation $x_{\ell-1} \in \mathbb{R}^{3 \times 3 \times 3 \times k_{k_{\ell}-1}}$, $f_\ell : \mathbb{R}^{3 \times 3 \times 3 \times k_{\ell-1}} \to \mathbb{R}^{k_\ell}$, and corresponding pseudo-label $z_\ell \in \mathbb{Z}$, we compute DB as: $DB = \frac{1}{k} \sum_{i=1}^{k} \max_{i \neq j} \frac{s_i + s_j}{d_{ij}}$ where $c_i$ is the centroid of the cluster $i$, $s_i$ is the cluster diameter and $d_{ij}$ is the distance between cluster centroids. A visualization of this metric with corresponding labeling is shown at each layer of the UNet in Figure 1, where each dot represents an intermediate $k_\ell$-dimensional pixel. In contrast, if

we were not using aforementioned receptive field analysis, each layer would yield a single dot that represents the layer's entire intermediate feature tensor (proportional in size to the input image volume). In either case, the intermediate features can be aggregated across many training images to develop a denser sampling of each node $f_\ell$ with more representative principle components and a better representation of clusters for computing DB.

In continuous-valued regression problems, the original or downsampled pseudo-labels do not take discrete values. In this case, a clustering can be achieved by histogramming and binning the output distribution, such that the discrete case where $z_\ell \in \mathbb{Z}^1$ is a special case of the general case $z_\ell \in \mathbb{R}^{k_3}$, even for multi-class groundtruth. See A.1.1 for an example.

### 2.3. Metrics without Labels

#### 2.3.1. ROUGHNESS

We first extend the cluster-based analysis using receptive fields and pseudo-labels to a label-free scenario by, instead, directly clustering the output space of layer $\ell$ and pulling back this cluster assignment to the input patches of $x_{\ell-1}$. To this end, we perform PCA ($d = 10$) and k-means ($k = 5$) on the output distribution $x_\ell \in \mathbb{R}^{k_\ell}$ of a large sampling of training data. (See A.2.1 for intuition behind parameter choices.) Then, using the same or a different set of data, $x'_\ell$, e.g. corresponding to different training/validation or test images, we project $x'_\ell$ to $\mathbb{R}^d$ using the learned components, subsequently assign each lower-dimension point to one of $k$ clusters. These cluster indices become the pseudo-"pseudo-label" for the corresponding input patch in $x_{\ell-1}$, thereby enabling evaluation of the DB index score or similar clustering metric. Specifically, we use clusters corresponding to the pseudo-labels derived from k-means at layer $\ell$ to measure the cluster separability or distortion of *image patches* at layer $\ell - 1$. Measuring separability in this way captures transitions of samples from one cluster group in the input space ($\ell - 1$) to another cluster group in the output space ($k_\ell$), serving as a simple measure of the gradient, or the node $f_\ell$'s "roughness", *in locale of the training data*. This is similar to the "sensitivity" metrics proposed in Novak et al. (2018). Figure 2 provides a visual overview of this technique, during both training and inference.

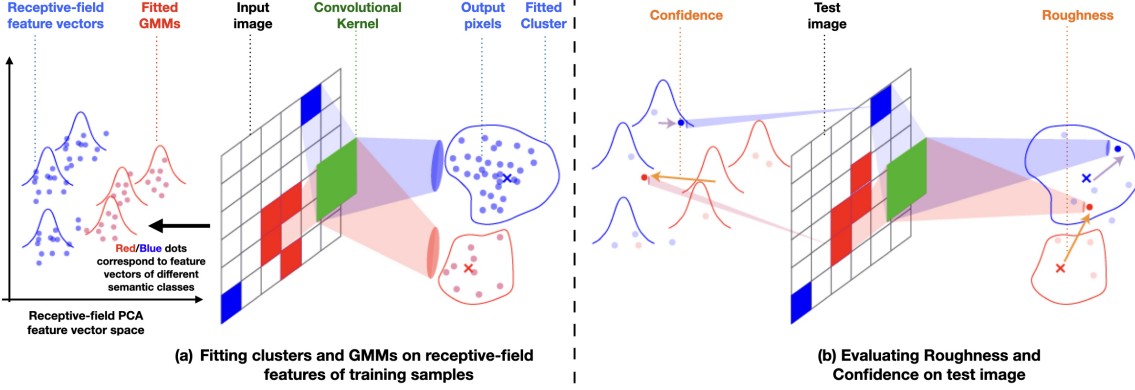

Figure 2: Illustration of the label-free roughness and confidence metrics computed, where output pixels at each layer are clustered to obtain "pseudo-labels" to measure the "roughness" of the layer with respect to the receptive field inputs.

### 2.3.2. CONFIDENCE

To more directly measure the intuition that well-performing networks with good generalization performance converge to dense representations on their nodal functions, we use the local feature manifolds to compute the density of the training data in the vicinity of new data. To achieve this, we train ($m = 3$) component GMMs on each of the k-means ($k = 5$) clusters computed on the dimensionally-reduced ($d = 10$) output representation of the training data $x_\ell$ at layer $\ell$. The use of GMMs enables density estimates that are better conforming to the geometry of the local feature space. With the PCA and GMM components fixed with parameter set $\Theta$, we then (*without labels*) evaluate the data likelihood of unseen data, $x'_\ell$, belonging to *any* of the mixture models regardless of the class-label or pseudo-class label after computing its low-dimensional projection $w'_\ell$, by computing a confidence value $c \leftarrow \max\{p(w'_\ell|\Theta_{\ell,m_j,k_i}) \ \forall \ m_j \in m, \ k_i \in k\}$. We multiply $c \in \mathbb{R}^1$ by the norm of the output pixel $x_\ell \in \mathbb{R}^{k_\ell}$, yielding a weighted-confidence for each input image patch at layer $\ell$. To compute a model-wise confidence for a given dataset, we take the mean over a large sampling of image patches. This effectively measures the average likelihood of *relevant* (e.g. ReLU- or sigmoid-activated) image patches belonging to the training dataset.

## 3. Experiments

### 3.1. Description of Prostate Segmentation Dataset and Trained Models

We use a dataset of 973 3D T2-weighted prostate-MRI exams, with corresponding radiologist-annotated or certified 3D annotation of the prostate region. The dataset was split into 80%-training and 20%-testing. A residual 3D UNet architecture $f^*(x)$ with group normalization ($g = 4$), operating on single-channel [64,64,16] patches of T2-weighed MRI, was optimized using a groundtruth binary-valued segmentation $f(x)$ with the Adam optimizer (learning rate 0.001) using one of 4 different objectives (Appendix A.2.2-A.2.3): (1) mean-square error, (2) class-balanced binary cross entropy, (3) Dice Loss, and (4) a combination of Dice loss and class-balanced binary cross entropy. For each objective, the model was trained from scratch with random initialization for 32 epochs each, resulting in a model dataset of 128 checkpoints that we subsequently use for our generalization prediction experiments.

The test accuracy for each model was evaluated using the intersection-over-union (IoU) metric for the full 3D exam, by assembling predictions on each patch into the original encoding space of resolution [0.313 mm, 0.313 mm, 2.4 mm]. It is worth noting that the absolute accuracy of these methods is not as important achieving a spread of IoUs that we can use to benchmark our measures of generalization. As such, we do not use validation data for early stopping, since we evaluate performance of every checkpoint on the test set.

### 3.2. Evaluation Procedure

For each model checkpoint ($m = 128$), we instrument the 3D UNet at each layer to extract features corresponding to training and testing images (T2w-MRI). Using these features, we compute the aforementioned label-dependent and label-independent metrics at each layer and take the average, weighted by the Frobenius norm of the model weights at each corresponding layer. The Frobenius norm helps to scale the layer-wise metrics relative to the dimension, sparsity, and energy of the representation at each layer (Liao et al., 2018). This results in a single scalar for each metric for each model, which we can use to evaluate a

model's generalization performance (test-set IoU) using the Pearson correlation measure $\rho$. We perform this analysis in two ways: (a) using the feature tensors at each layer as a single feature vector, and (b) using the local receptive field analysis presented in this paper. For the DB clustering metric we evaluate on a random subset of $n = 198$ training exams with annotations, whereas for roughness and confidence we evaluate on $n = 198$ testing exams without annotations. In all cases, no test-set annotations were used to compute the metrics. Due to memory constraints, a random subset of 80K extracted features vectors are used.

### 3.3. Correlation Across Models

As shown in Table 1 and Figure 3, for clustering (DB), roughness, and weighted confidence metrics, the PCA-based methods using local receptive fields outperforms methods that interpret the tensor images as a single feature vector. The absolute value of the correlation coefficient $\rho$ indicates how well the metric correlates with the IoU of the test set. The *label-free* confidence metric on test data performs the best, but surprisingly has a negative correlation factor. This behavior is also demonstrated in other CNN models (See A.1.2).

|  | DB on Train Data | Roughness on Test Data | Confidence on Test Data |
|---|---|---|---|
| Original PCA Method | -0.600 | -0.629 | -0.651 |
| PCA with Local Receptive Fields | -0.704 | -0.783 | -0.883 |
| Absolute Percent Improvement | 17.3% | 25.5% | 35.6% |

Table 1: Pearson correlation coefficient $\rho \in [-1, 1]$ of different methods and metrics evaluated on the dataset of $m = 128$ model checkpoints and $n = 198$ 3D MRI exams.

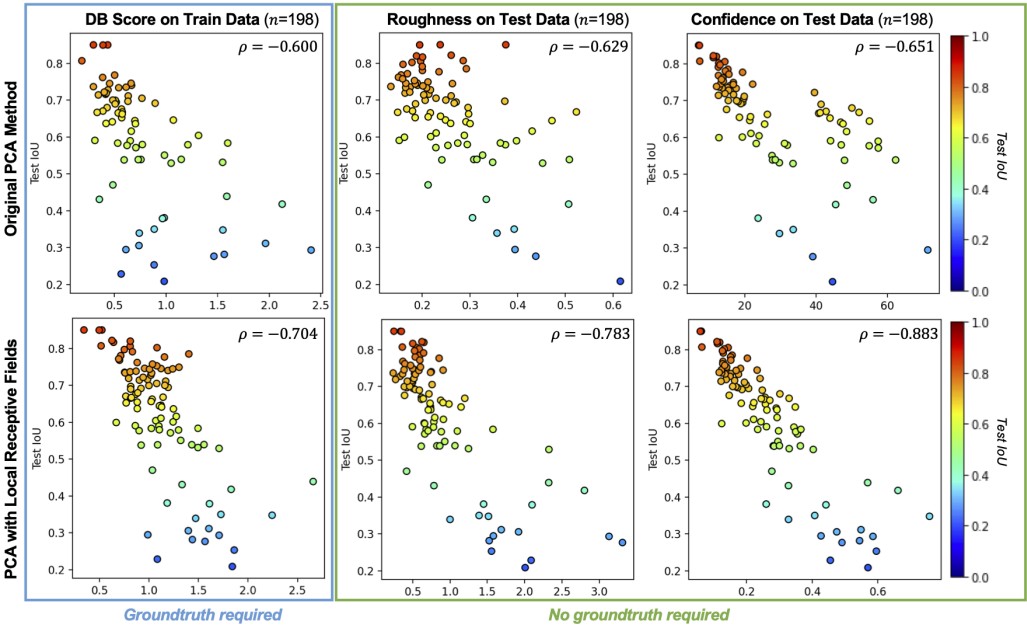

Figure 3: Visual comparison of different measures of generalization and how they correlate with the test IoU (y-axis and colorized) for the prostate segmentation models.

Figure 4 visualizes this trend as a function of layer, demonstrating that the proposed receptive field analysis results in a better striation of models (colors) over layers. Interestingly, the weighted confidence score has a local minima in layer 7, the fully-encoded portion of the UNet, complementing the observation of untangling near the base of Figure 1.

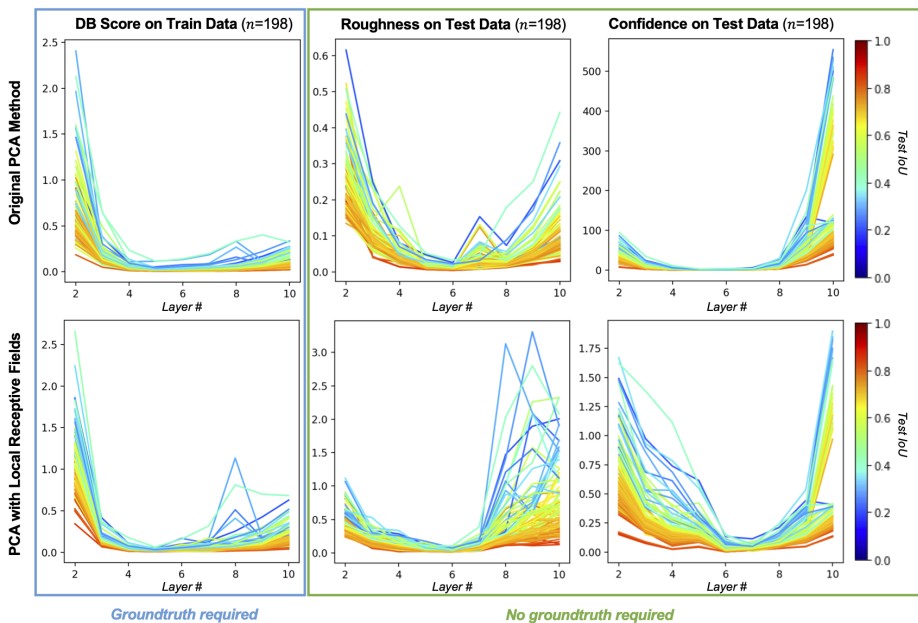

Figure 4: Comparison of how different measures (y-axis) of generalization correlate with the test IoU (colorized) across layers (x-axis) of the prostate segmentation models.

## 4. Discussion and Conclusion

The key insight of this work is to utilize the compositional, convolutional structure of the UNet to better understand and visualize the neural network "trace" of an input query image. Although the presented metrics are far from a perfect score ($|\rho| = 1$), the presented receptive field analysis provides a computationally-tractable technique to analyze the generalization capabilities of UNets trained with arbitrarily-sized inputs, which may be utilized in future metric discovery. The low dimensional compressed representation of nodes is also theoretically pleasing because it brings modern CNNs closer to our understanding of compositional networks, and serves to better explain their strong generalization capabilities. While this paper has focused on 2D/3D fully-convolutional networks, the same analysis applies to other feed-forward CNNs and DNNs, although additional innovations are required for recurrent and multi-branch networks, or to better capture other sparse-matrix operations.

We note that the presented methodology differs in spirit from "supervised" techniques, which attempt to discover data-driven metrics that predict generalization, either during training or using an additional DNN that learns a metric for generalization (Unterthiner et al., 2020). Still, care must be taken to ensure such "unsupervised" generalization metrics are not overfit to particular datasets, objective functions, and optimizers, even if the architecture is fixed. In this vein, further theory-based analysis and insight into the local geometry of features is required, e.g. to better understand the negative correlation factor for the weighted confidence metric observed in CNNs, and now in UNets. Utilizing theory-based approaches to understanding generalization will not only reduce the search space, but may provide inspiration for new architectural or objective function innovations to improve robustness of networks in critical applications such as in medical imaging and recognition.

## Acknowledgments

This research was supported by NIH/NIBIB grant #F32EB030411 and NIH/NCI grant #R01CA229354.

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

# Appendix A. Supplementary Material

## A.1. Results on Convolutional Neural Networks for Image Classification

### A.1.1. Limitation of Clustering Metrics

In this section, we take a look at the performance of clustering metrics proposed for predicting generalization of image classification CNNs in NeurIPS PGDL Competition 2020, where the clustering is done on interior features using corresponding image labels. We observed that the reason for euclidean based clustering metrics like DB index correlating better with generalization accuracy is due mainly to its better performance on the fully-connected (FC) layers. Visualizing the clusters revealed that the clusters are very well separated for FC layers, for example with Layer 9 in Figure 5(b) which is a FC layer, in contrast with Layer 7 which is a convolutional layer. We attribute following reasons for the difficulty in extending these clustering metrics to convolutional layers: 1) The dimension of a convolutional layer's feature vector for the entire image is very large compared to that of a FC one and so the euclidean based metrics overestimate the distances in such high-dimensional spaces, 2) Fully connected layers pool the information from all the channels of the convolutional layers and are thus able to correlate their feature vectors better with the image labels, whereas the receptive field of a convolutional layer only corresponds to a part of the image and so clustering its feature vector using the image label may not be appropriate. Due to these reasons, the metrics developed primarily for classification CNNs, which in most cases employ FC layers at the end, cannot be extended directly to the study of fully convolutional models. Figure 6 shows this disparity between VGG (Simonyan and Zisserman, 2015) like models with FC layers and fully convolutional models like Network in Network (Lin et al., 2014) from the PGDL dataset. Contrast this with Figure 4 and Figure 7(b) which uses our receptive-field analysis on convolutional layers producing more robust metrics for better understanding the fully convolutional models.

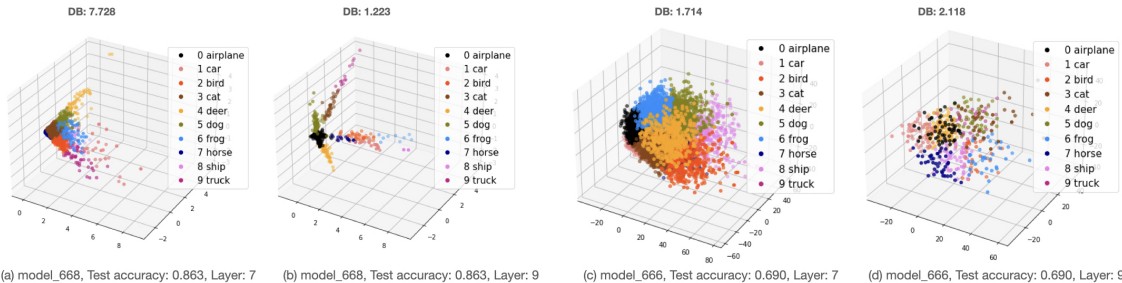

Figure 5: Clustering of the intermediate feature images (visualized in 3D using PCA) for two NeurIPS PGDL image classification models, with associated Davies-Bouldin (DB) scores. In some layers, the model with higher test accuracy (`Model 668`, acc. 86.3%) has a higher DB score than the model with poor generalization (`Model 730`, acc. 69.0%), demonstrating that class separability of intermediate feature images is often misleading.

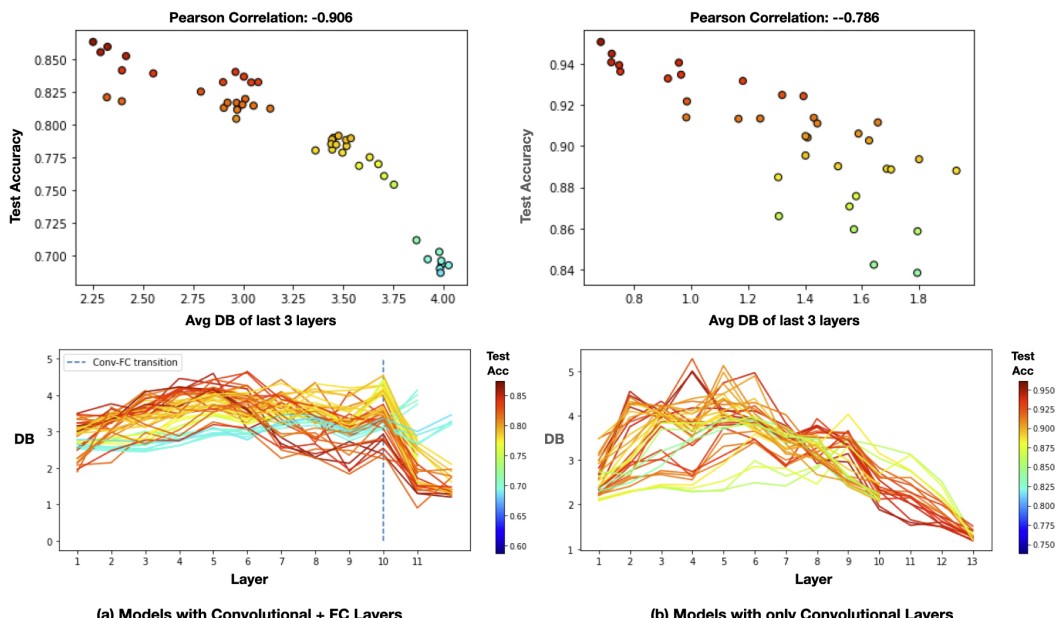

Figure 6: (Top) Correlation of Average DB score (without receptive-field) of last 3 layers with test accuracy for models from NeurIPS PGDL dataset. (Bottom) DB Score plotted for each layer of these models. DB metric without receptive-field clustering for convolutional layers is unreliable and the overall metric performs poorly as a generalization evaluation metric for fully convolutional models. Models in (a) are chosen from Task 1 of PGDL dataset which have more than 10 layers. Models in (b) are chosen from Task 2 with depth=12.

### A.1.2. Evaluation of Confidence Metric on Classification CNNs

We evaluated our proposed method, Confidence without labels using receptive-field analysis on PGDL dataset's classification models. Figure 7(a) shows the performance of the model in predicting the generalization of these models achieving a Pearson correlation coefficient of -0.890. Moreover, our method is consistent in evaluating the convolutional features as well as fully connected ones, across all the layers of the model, as shown in 7(b).

## A.2. Implementation Details

### A.2.1. Choice of Generalization Metric Hyper-Parameters

**Number of clusters $k$**

In this section we provide the reasoning for the selection of parameters like number of PCA components, number of clusters and number of components in GMMs. We first note that in the local receptive-field analysis, the labels for the feature vectors at interior layers do not necessarily correspond to the neural network's output classification labels, but rather approximate the activations of the kernel weights corresponding to each receptive field. To this effect, the choice of number of clusters cannot be directly obtained from the total number of classes when clustering the receptive field feature vectors. For example, a

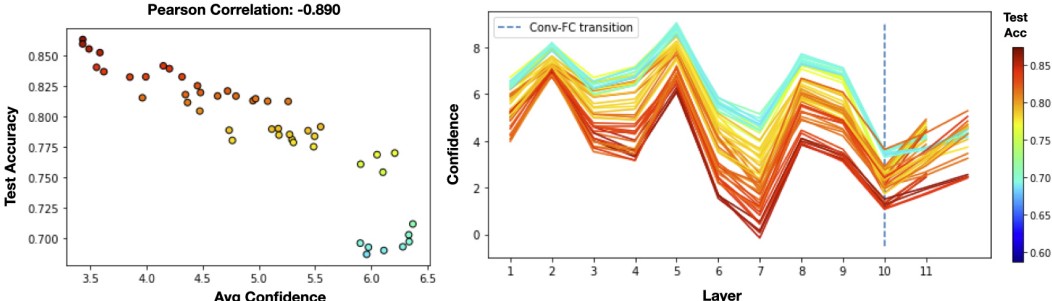

Figure 7: (left) Correlation of average confidence with test accuracy for models with more than 10 layers from Task 1 of NeurIPS PGDL dataset. (Right) Confidence score plotted for each layer of these models. Confidence metric with receptive-field clustering outperforms DB on this subset of classification CNNs.

receptive field patch of size 7x7x64 may correspond to the patch of grass in the background and ideally we would want the activation of this patch belong to a cluster of similar patches from input images belonging to any output class. In Figure 8 we plot the clusters with varying number of cluster centers $k$ for features from Layer 1 of a CNN model (`Model 600`) from the PGDL dataset. For $k = 10$ (3rd column) we see that the cluster labels assigned are not well separated in the euclidean space. On the other hand for $k = 3$ (first column), points which are "far away" from the "dense arms" of the cluster centers are also assigned same labels. So we chose $k = 5$ for our experiments which seems to be doing a good job assigning a different label to points which lie away from the dense clusters while separating the clusters well in this euclidean feature space. We fixed $k = 5$ for all convolutional layers for the local receptive-field analysis. However, the activations of deeper convolutional layers may be different from those of shallow ones which might result in better clustering if we choose a different $k$ for different layers. But this will impose additional computational burden to do such hyperparameter optimization dynamically, so we plan to explore this in our future experiments. But we also note that our receptive-field analysis provides a visual guide for such hyperparameter selection instead of requiring to do full grid search for best performance.

**Number of PCA components**

We have also performed experiments to observe the effect of PCA dimensionality reduction on the computed metrics. Table 2 shows the variation of the average clustering metrics `Roughness` using DB index score and the likelihood metric `confidence` for `Model 600`. Figure 9 shows the variation of these scores for all layers for different values of number of PCA components `pca_comps` and also without any PCA dimensionality reduction. Choosing a very small value for `pca_comps` will throw away a lot of information from the receptive-field features and the corresponding metrics obtained could not be relied upon. Choosing `pca_comps` above a certain threshold stabilized the confidence metric revealing that above this threshold value, the extra dimensions do not offer any further information on the quality of the clustering when measured using likelihood, while significantly reducing the computational time from 37.5 min for clustering without PCA to 51s for `pca_comps = 10` while also enabling us to visualize the clusters. Although the DB score approximates the

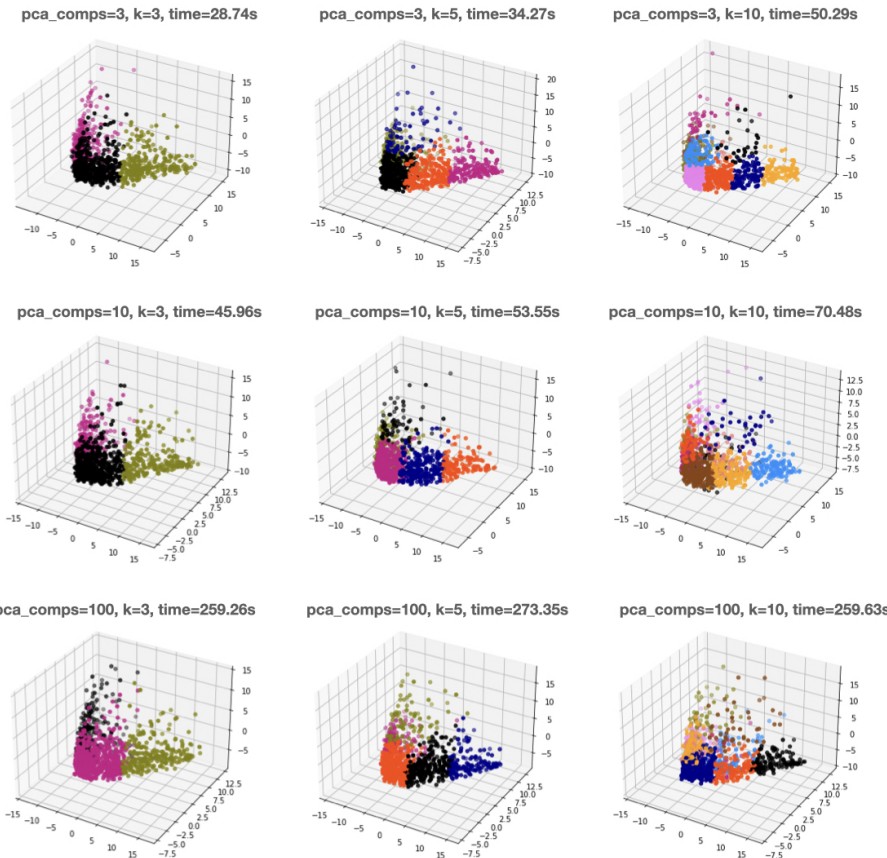

Figure 8: Clusters of the PGDL challenge `Model 600`'s Layer 1 receptive field features plotted for different values of Number of PCA components (`pca_comps`) and Number of cluster centers (k). Cluster labels with k=3 cannot separate out-liers from in-liers, while with k=10 forcibly assigns different labels for points even which are packed together. Increasing the `pca_comps` has little effect on cluster quality but significantly increases the computation time.

roughness well for clusters with and without PCA reduction, it is clear from the figure that this metric is affected by performing PCA on the features, a shortcoming of euclidean based metrics on CNN feature spaces which we discuss further in the next section on GMMs. Thus we select `pca_comps` = 10 for our experiments which is a good compromise between computational time and preserving the higher dimensions needed for a reliable computation of the confidence metric. Looking at the columns of Figure 8 from top to bottom also gives us a visual confirmation that choosing `pca_comps` = 10 for PCA dimensionality reduction does not significantly affect the quality of clusters formed compared to those with higher dimensions.

**Number of GMM components**
Choice of number of GMM components also follows a similar reasoning as that of number of cluster centers $k$. Since the GMMs are used to obtain the density estimates, we try to choose the number of components based on the geometric spread of the cluster, allowing

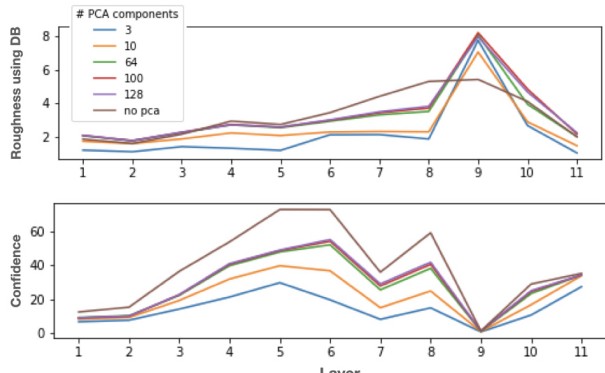

| No. of PCA Components | Roughness | Confidence | Time(s) |
|---|---|---|---|
| 3 | 2.148 | 14.516 | 34.5 |
| 10 | 2.515 | 21.453 | 53.9 |
| 64 | 3.179 | 27.641 | 179.7 |
| 100 | 3.334 | 28.513 | 305.6 |
| 128 | 3.32 | 28.953 | 409.9 |
| No PCA | 3.25 | 38.71 | 2516.6 |

Table 2: Average Roughness, Confidence and computation time for `Model 600` with and without PCA.

Figure 9: Roughness(top) and Confidence(bottom) at each layer of `Model 600` for different number of PCA components and also without PCA.

for points at the boundaries in each cluster to have equal likelihood representation as that of points at the center of the clusters. This enables us to NOT rely upon measuring the euclidean distance of the receptive-field features from the trained cluster centers during validation, which is known to be extremely unrepresentative in the neural network feature space for measuring the similarity between the computed features, thus giving us a more robust confidence measure. So choosing a single GMM component only offers minimal advantage over computing the euclidean distance from cluster centers, while choosing a large number of components can lead to a false confidence when it can overfit even out of distribution receptive-field features from the validation set, as can be seen from Figure 10).

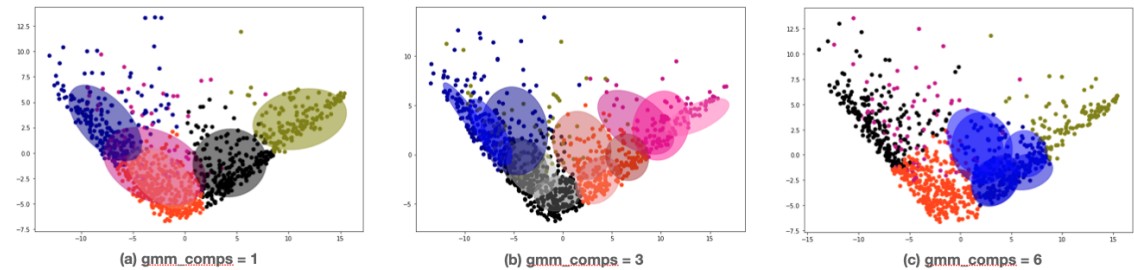

Figure 10: Covariances of GMMs for different number of components fitting the clusters. 3 components fit better for most of the clusters (Covariances of only 1 cluster are shown for `gmm_comps`=6 for clarity). Ellipses of same color with different gradients correspond to multiple components of the same mixture.

### A.2.2. 3D Residual UNet Architectural Parameters

For all experiments we use a depth-5 3D residual UNet (sometimes called 3D VNet) with $3 \times 3 \times 3$ kernels with stride 2, where depth corresponds to the number of "stages" (convolutions + downsampling in the encoder, convolutional + upsampling in the decoder), and not necessarily the number of arbitrarily-defined "layers" (e.g. group-norm, etc.). This results in a network with 11 layers. We utilize outputs from 9 of those layers for our analysis (Layer 2 to Layer 10).

### A.2.3. Training Objectives

We train models with one 4 different standard objectives for semantic segmentation:

1. Mean-square error: $\frac{1}{n} \sum_n ||f(x) - f^*(x)||_1$

2. Class-balanced binary cross-entropy (BCE): $\frac{1}{z} \sum_z \frac{1}{n_z} \sum_{x, f(x)=z} \text{BCE}\big(f(x), f^*(x)\big)$

3. Dice loss (Milletari et al., 2016): $\left(1 - \frac{1}{n} \sum_n \frac{2 \cdot \sum f(x) \cdot f^*(x)}{\sum f(x)^2 + \sum f^*(x)^2}\right)$

4. Combination of Dice loss and class-balanced binary cross-entropy loss:
$\left(1 - \frac{1}{n} \sum_n \frac{2 \cdot \sum f(x) \cdot f^*(x)}{\sum f(x)^2 + \sum f^*(x)^2}\right) - \log\left(\frac{1}{z} \sum_z \frac{1}{n_z} \sum_{x, f(x)=z} \text{BCE}\big(f(x), f^*(x)\big)\right)$

### A.2.4. Checkpointing and Model Evaluation

Four identical models are trained using the 4 aforementioned objective functions, respectively, for 32 epochs each. We save the model weights at the end of each epoch, resulting in $4 \times 32 = 128$ model checkpoint files. The test accuracy (IoU) file is evaluated for each model separately on the entire test set in 3D. Specifically, several [64,64,16] pixel image patches are evaluated and assembled to the form the full resolution prediction for each MRI exam in the test set. The IoU score is computed by comparing the predicted volume (quantized to $\{0, 1\}$ using a voxel threshold of 0.5) to a groundtruth binary segmentation mask annotated by an expert radiologist. All model checkpoints are evaluated equally, with no special consideration given to any epoch # nor objective function used.

For evaluation of the metrics, dataloaders corresponding to the training and testing data were generated in order to sample images batches that are evenly distributed across the whole dataset. In particular, a sampled batching strategy is used to return [64,64,16] pixel patches from the semantic groups: (1) 0% of patch volume is prostate gland, (2) $\leq$ 15% but $> 0\%$ of patch volume is prostate gland, and (3) $\geq 15\%$ of the patch volume is prostate gland. This was done to ensure adequate sampling of gland, non-gland, and gland-bordering regions of the prostate.

### A.2.5. Source Code

Available here: https://gitlab.com/abhe/UNet-Generalization_MIDL2021.

