# OpenReview forum: "Understanding and Visualizing Generalization in UNets"
_MIDL.io/2021/Conference — MIDL 2021_

### Official Review · AnonReviewer4 · 2021-03-05

**Confidence:** 2
**Preliminary Rating:** 2
**Final Rating:** 3

**Summary:**

The paper proposes to understand and visualize the generalization performance of U-nets without ground-truth annotations. The proposal includes dimensionality reduction at the receptive field level, k-means clustering to create pseudo-labels, and GMM to estimate the likelihood of new data. Empirical studies were carried out on T2-weighted prostate-MRI and the results were claimed to show some degree of correlation with model generalization.

**Strengths:**

1. Visualizing the generalization of U-nets is an important area of research and the paper highlighted the drawbacks of existing approaches.
2. The efforts towards label-free generalization metrics and visualizations are challenging and very much needed.


**Weaknesses:**

In general, I think this paper is hard to follow and lacks sufficient empirical results to support the main claims.

1. I find it difficult to understand how "Instrumentation and Local Receptive-field Analysis" is designed exactly.
The paper attempts to formalize a layer into a mapping function from $R^{3\times 3\times 3 \times 1}$ to $R^k$ where $k$ represents the number of feature channels, where $k$ is further reduced to $d$ through PCA. I don't understand why and how the receptive field can be reduced to $k$ in the first place. How it differs from directly applying PCA to the receptive field to $d$ dimensions without the step in between.

2. The paper lacks empirical evidence to support the proposed approaches.
The first proposal in the paper, Instrumentation and Local Receptive field, should be evaluated against methods without local PCA. The same with the full model in Section 3.2. The lack of comparison makes it very difficult to know how effective the proposal is. Furthermore, the correlations in Fig. 5 are too weak to draw meaningful conclusions.

**Deanonymize Review:**

no

**Final Rating Justification:**

After re-reading the paper and related work, I think this paper has some novel contributions to interpreting the U-nets; therefore, I upgrade my score to "Weak Accept."

**Justification Of The Preliminary Rating:**

Although label-free generalization analysis of U-nets is a good research direction, this paper is very hard to follow and lacks sufficient empirical results and ablation studies to support the main claims.

**Paper Type:**

methodological development

**Questions To Address In The Rebuttal:**

Please see above.

**Special Issue:**

no

---

> ### Author Response · Authors · 2021-03-18
> **AnonReviwer4**
>
> We thank AnonReviwer4 for their helpful and constructive comments. We understand the paper could have used better organization, so we have made several updates.
>
> Weaknesses:
> 1. The local receptive field analysis is crucial to this paper, so we have made revisions to hopefully clarify how this is performed.  The receptive field analysis reduces the image-to-image operation of a UNet stage (which, in the encoder, performs convolutional+downsampling) to multiple image-to-vector operations. In the conventional PCA approach, the input/output relationship is not used; at a given layer L, the (D_1 x D_2, x D_3 x k_L) feature images at layer L are compressed to “d” dimensions using PCA. Due to such high compression ratios, a lot of information may be lost. (Note: “d” is kept small to allow computational tractability and also increase performance of clustering.) In contrast, the receptive field analysis has two parts. First part is to identify which input images contribute to which output pixels. For example, in the encoding branch of the UNet, the input “image” to layer L is always a 3x3x3xk_{L-1}) patch at layer L-1, regardless of the image size. The output/result of applying the function at layer L, f_L, to that input patch results in a k_L sized vector (at a corresponding image-pixel location). Now, performing PCA we only need to compress the feature channels k_L to d dimensions, rather than the whole image! The second part is to associate each of those k_L or d-dimensional vectors with patches/clusters at layer k_{L-1}, which gives rise to the clustering, roughness, and confidence metrics discussed in the paper.
> 2. We have now updated the evaluation section with comparisons across existing PCA methods, which are shown to perform worse for UNets due to a low data rate and high computational cost.
> We have also added more checkpoints from additional training, which have resulted in slightly higher correlation scores for the proposed generalization metric.

---

> > ### Comment · AnonReviewer4 · 2021-03-18
> > **Thanks for the clarification**
> >
> > I would like to thank the authors for clarifying "Local Receptive-eld Analysis." After re-reading the paper and related work, I think this paper has some novel contributions to interpreting the U-nets; therefore, I upgrade my score to "Weak Accept."
> >
> > I strongly recommend the authors improve the writing of section 2.1 with more clarity and explanations regarding why traditional PCA would fail on segmentation tasks. In particular, I think the prior work on PCA obtained from feature maps are too high granular to provide sufficient information for segmentation tasks---please correct me if this is incorrect. I am not convinced the challenge is computational, and I think it might be more related to the task being segmentation.
> >
> > The authors should also comment on whether the proposed local approach is designed for segmentation tasks exclusively or could potentially be used in classification tasks as well. The potential concern for using local approaches for classification is that many receptive fields are not contributing to the classification task, which might introduce significant noise into the interpretation framework. I understand this is outside of the scope of this work, just wanted the authors to comment on this to help the community with future directions.

---

### Official Review · AnonReviewer2 · 2021-03-08

**Confidence:** 5
**Preliminary Rating:** 4
**Recommendation:** Oral
**Final Rating:** 4

**Summary:**

The authors take a deeper look at the ubiquitous convolutional neural networks.
They acknowledge that often little labeled data is available, and metrics on small validation datasets might not reveal how well a model works.
Therefore discuss existing methods in detail and develop new ones, to investigate internally how a specific Unet works, and estimate how well it can generalise.
They develop a measure that can be evaluated on un-labeled data.
By correlating this measure with actual validation scores, they find a (albeit not very strong) correlation.

**Strengths:**

The problem the authors tackle is extremely relevant, and the paper is well written and embedded in literature.
The paper is quite varied and gives many interesting pointers to find different angles to evaluate and develop new models, past the simple 'show performance on a specific validation dataset'.
These methods are very relevant to real applications of convolutional networks, where labeled data is sparse, and the variation in data is large, thus estimating generalisation capabilities are very valuable.

**Weaknesses:**

The final metric does not have a very strong correlation (as acknowledged by the authors).
Although the correlation metric itself is already interesting, and a possible metric for future unsupervised 'generalisation metrics' to optimise.

The discussion of existing metrics by previous work, and what is newly introduced by the authors, can be a bit hard to separate. Part is discussed under 1.1 (prior work), but it continues in all further subsections. Although contributions are more clearly spelled out under 1.2 (contributions).

**Deanonymize Review:**

no

**Detailed Comments:**

Many lines that start with a citation have the citation between brackets: "(Zhang et al. 2018) introduce...". I'm not sure if the style allows it, but more commonly you use a different citation command to make the citation  in the sentence like: "Zhang et. al. (2018) introduce...".

The description of the Figures, especially 5, could be more self-contained. There are several axes and the labels are coloured. So this is quite some information, which might need a bit more explanation.

**Final Rating Justification:**

I thank the reviewers for addressing the comments. The figures also look much better now.

**Justification Of The Preliminary Rating:**

The subject matter is very relevant for the field, that is dominated by many incremental changes to models, instead of looking at more innovative approaches to evaluate such models.
The ideas are collected together, so this paper could form a good point of inspiring more of such methods.

**Paper Type:**

methodological development

**Questions To Address In The Rebuttal:**

Are there some heuristic used to determine the parameters for the PCA and k-means in the Roughness estimation?


**Special Issue:**

yes

---

> ### Author Response · Authors · 2021-03-18
> **Response to AnonReviwer2**
>
> We thank AnonReviwer2 for their helpful and constructive comments. We are glad the importance of this problem is clear! We have added additional clarifications and updates per your comments.
>
> Weaknesses:
> - We have updated our correlation results using additional checkpoints from additional training, an updated weighted confidence measure, and are now using the Pearson correlation coefficient for clarity. This has resulted in higher correlations values. We have also added plots depicting this correlation across the various layers of the network.
> - We have now added a comparison to the previous analysis methods, which treats each feature tensor as a single, large feature vector. Our results demonstrate that the local receptive field method is superior. We have sequestered discussion of existing literature to the Prior Work section.
>
> Detailed Comments:
> - We have updated the text with the correct citation style.
> - We have updated the scatter plots and caption to better indicate the meaning of the axes, colors, and trends, including comparisons to the conventional approach that performs worse.
>
> Questions:
> - We have added Supplementary Material (Appendix A.2.1) which discusses heuristics and rationale for choosing PCA, k-means, and GMM parameters.

---

### Official Review · AnonReviewer1 · 2021-03-08

**Confidence:** 2
**Preliminary Rating:** 1
**Final Rating:** 3

**Summary:**

The paper proposes techniques for characterising and visualising the generalisation of UNets in image classification and regression. The authors develop metrics that perform these evaluations in a label-free setting, proposing to alleviate current reliance on ground-truth annotations. Specifically, they extend cluster based analyses to the visualization in the fully convolutional setting (i.e. U Nets), with their label free metrics correlating with test generalisation.

**Strengths:**

The paper seems to tackle an important question in deep learning- that of providing visual/ quantitative notions of generalization beyond simply a test/validation performance that examines intermediate representations within deep networks. Although the models being examined in the paper may be limited to 3D U nets, the principles being introduced seem to be general and may warrant broad interest.

**Weaknesses:**

1. To me, the paper was a bit confusing to follow, with many details either lost within the explanation or the exact setup often lacking in explanation/justification and intuition.
2. Within the introduction of metrics (section 2.2), the authors do not provide enough intuitions for the extensions they propose, or theoretical justification for correctness, or extensive evaluation under different test scenarios.
3. The authors neither mention clearly nor justify why certain modeling choices were made during evaluation (for example, number of dimensions in PCA, number of clusters in k-means, number of mixtures in GMMs).
4. The experimental scope is also limited in terms of replicability of the results. Specifically, only one dataset and one U Net architecture is evaluated. Neither are performance trends evaluated for consistency within the dataset by subsampling/bootstrapping. Although different loss functions for training were used as a proxy for different models, the results overall are not convincing enough in terms of the main thesis of the paper.

**Deanonymize Review:**

no

**Detailed Comments:**

Clarifications:

1. Section 2: What do Model_668 and Model_666 refer to? Why are they being compared? When the authors refer to inconsistencies in terms of clustering within intermediate layers, is it possible that treating outputs of a convolutional layer the same way as those of a fully connected one give rise to this effect, or perhaps the use of PCA for visualisation-when linear dimensionality reduction may not be appropriate? I'm not sure what the takeaway from this section is in the context of the paper.
2. What is the effect of downsampling of the labels in section 2.1, especially at the segmentation boundaries?
3.  Subsection 2.2: Without loss of generality, we assume that ... and not necessarily the number of arbitrarily-defined “layers” (e.g. group-norm, etc.) - what do the authors mean by arbitrarily defined "layers" (e.g. group norm, etc)?
4. Subsubsection: 2.2.2: What do the authors mean by adequate clustering. In general, it seems non-trivial to view a regression problem as a classification/clustering problem, since the clusters have an inherent ordering. Needs better justification
5. Subsection 2.2.3: Mixup: x_1 + \alpha x_2 should not strictly be a convex combination unless \alpha is 0. In general, would taking the convex combination of medical images within a population be an appropriate data-augmentation strategy, given individual differences?
6. Subsection 2.3.1: "Then, using the same or a different set of data, x0' , e.g. corresponding to different training/validation or test images ..."- does this mean that the entire data is used to infer a PCA basis? Why?
7. Subsection 2.3.1: "This method effectively provides a measure of the quantized gradient, or “roughness”, at layer l in locale of the training ... "- this is not obvious why



**Final Rating Justification:**

The authors have addressed all the points that I raised during the initial review and have made substantial changes to improve the readability of the text. Upon examining the revision, I'm willing to update my initial rating to a weak accept as I now agree that the methodological contribution is sufficient for a broader audience and is much better justified and explained. Lastly, I would invite the authors to include a discussion on the computational bottleneck for using traditional PCA for segmentation within the paper (preferably within the main text).

**Justification Of The Preliminary Rating:**

To me, the presentation of the paper was not clear, results not convincing, and too many points unexplained within the methods and experimental setup. Overall, I think this paper is below the acceptance threshold.

**Paper Type:**

methodological development

**Questions To Address In The Rebuttal:**

Along with the aforementioned, within the results section, could the authors also clarify:

1. What do the colours in Fig. 5 correspond to?
2. What were the accuracies of each models?
3. How were parameters for computing evaluation metrics (PCA, GMM, k-means) selected and why?
4. Why are (weighted) averages at each layer taken?
5. "Similarly, models with higher overlap between the distribution of train and test data on interior layers, have higher confidence, and tend to have higher generalization performance" How is this to be inferred from the plot?


**Special Issue:**

no

---

> ### Author Response · Authors · 2021-03-18
> **Response to AnonReviwer1**
>
> We thank AnonReviwer1 for their detailed and constructive comments. We understand the paper could have used better organization, so we have made several updates.
>
> Weaknesses:
> 1. To improve the flow and focus of the paper, we have sequestered results on PGDL to the Prior Work section, and included references to Appendix A.1 that now includes a discussion of existing metrics, results, and their limitations.
> 2. We have added Appendix A.2, where we have added our intuition and justification of heuristics used to choose the PCA, k-means, and GMM parameters.
> We have also added a comparison of our results with existing methods, namely using extracted feature tensors as a single feature vector, demonstrating that the newly presented local receptive field analysis provides a strong boost in performance.
> 3. ^
> 4. For this initial demonstration we have focused on one dataset and architecture, which was chosen to introduce and clearly illustrate the methodology using a simple well-understood task. Given the ubiquity of Unets in medical imaging, metrics and initial demonstration on only UNets will still serve a broad audience. However, in Appendix A.1, we have added additional results on CNN classification models, including fully-convolutional models (albeit, not image-to-image networks).
>
> Clarifications:
> 1. Section 2: We have moved this Figure to Appendix A.1, where we include additional description of the models presented and what we mean by inconsistencies in terms of clustering. Although the PCA+clustering method struggles sometimes, even for fully-connected layers, its main drawback is in the convolutional layers. We attribute this to the large dimension of the intermediate feature (images).
> 2. Section 2.1: The effect of downsampling + ceiling is to overestimate the segmentation mask at the boundaries.
> 3. Subsection 2.2: The term “layer” in DNNs is ambiguous and has little meaning when many operations are strung together. Therefore, in this paper, we refer to the action of a “layer” to be a “stage” of a UNet. However, the proposed approach works equally well for characterizing any operation in a DNN. For example, for input x, the first layer can be simply y_1 = A*x, and the next layer can be y_2 = y_1 + b, followed by y_3 = sigma(y_2). Instead, we lump many of these operations together. Specifically, for Residual UNet, a single stage is defined as conv(conv(x) + x) + x. We have now clarified this point in Appendix A.2.2.
> 4. Section 2.2.2: We have clarified this text. We agree its non-trivial for arbitrary scales, but recent work has shown that DNNs can cut in the range space of the target function. Therefore, with sufficiently small radius, clustering in the output space can cluster similar samples in the input (or intermediate feature) space as well. This is particularly true when the gradient of the network is stable over the domain of interest, as has been shown for well-trained models. Specifically, the roughness measure captures when the gradient (Sobolev norm) of a network/layer is high, since this results in cluster-to-cluster jumps across the layer. We have added a figure to help clarify this.
> 5. Section 2.2.3: Correct, we meant \alpha_1 x_1 + \alpha_2 x_2, with \sum \alpha_k = 1 . Additionally, per aforementioned changes, we have removed this subsection since we do not believe Mixup is a viable strategy for training or evaluating medical images and UNets at this time.
> 6. Section 2.3.1: No, it is computationally challenging to use all of the data to compute the PCA. Therefore, we can use other training images (not used for PCA) or test images (definitely not used for PCA) to evaluate the metric. This effectively uses PCA to measure how well the feature images themselves generalize across different cuts of the dataset.
> 7. Section 2.3.1: We have clarified this in our comment above (Subsection 2.2), and have updated the text to reflect this (including in Subsection 2.3.1).
>
> Questions:
> 1. We have clarified the axes, colors, and caption of the scatter plots. The colors provide a visual representation of test accuracy (IoU).
> 2. The test IoU is now shown on the y-axis of Figure 3-4.
> 3. Appendix A.2.1 includes discussion of how to choose these parameters and why.
> 4. Weighted averages are taken to normalize the size/effect of each layer. It has been shown that the Frobenius norm, itself, is a measure of generalization in DNNs. We have added a citation with additional analysis. Averaging is used to reduce the layer-wise metrics into a single number used to predict generalization, although more complicated techniques (and even supervised learning techniques) could be used.
> 5.  The clustering and roughness plots have a negative correlation. This is also true for the weighted confidence measure. We have updated the caption and text to clarify this.

---

### Official Review · AnonReviewer3 · 2021-03-09

**Confidence:** 3
**Preliminary Rating:** 3
**Recommendation:** Poster
**Final Rating:** 4

**Summary:**

This paper proposes a method to evaluate the generalization performance of fully-convolutional neural networks (f-CNN), particularly Unet-based models, in image classification and regression tasks. Three contributions are claimed: extension of the feature-based analysis of DNNs for classification tasks to f-CNN for image-to-image tasks, an improvement of conventional cluster-based analysis, a new metric of generalization that does not require image label data.

**Strengths:**

* The paper is generally easy to follow.

* It addresses the problem of existing method in evaluation of generalization of neural networks.

* It proposes an interesting method to evaluate f-CNNs which includes computation cost and better exploits the structure of the network


**Weaknesses:**

* Experiments are limited only one type convolutional neural network: UNet. The paper would be strong if its analysis extended to more f-CNN architectures.

* The experiments section is short and lacks sufficient details to fully understand the proposed analyses.


**Deanonymize Review:**

no

**Detailed Comments:**

*  In section 3.1, the authors should provide more details about the experiment setting. For example, the four different objectives are mentioned, but it is not clear how they were used in the training? Are they used to train different networks separately or in jointly?

* Why only split the dataset in train and validation set, instead of the standard train/validation/test sets?

* How did the authors obtain 120 checkpoints for analysis?

*Authors claim their method improves the existing approaches, however there is no comparison in experiments. This weakens the claims.


**Final Rating Justification:**

*  In the rebuttal and the updated manuscript, the authors addressed most of my concern. I appreciate the authors’ effort to update the manuscript. The additional details in the manuscripts and the appendix have made the paper stronger and illustrate the paper’s idea clearer.

*  In this version, the authors also have made the paper's source code public, which is very helpful to reproduce the result and reuse them for other works.



**Justification Of The Preliminary Rating:**

The paper presents an interesting analysis which can help understand the generalization ability of f-CNNs. However, this analysis is limited to a single f-CNN architecture. Moreover, experiments lack details and do not compare against alternative/current approaches.

**Paper Type:**

both

**Questions To Address In The Rebuttal:**

* Test other f-CNN architectures or explain why U-Net is sufficient.

* Add more details about experiments.

* Provide empirical evidence of the proposed method's advantage compared to existing approaches.

**Special Issue:**

no

---

> ### Author Response · Authors · 2021-03-18
> **Response to AnonReviewer3**
>
> We thank AnonReviwer3 for their helpful and constructive comments. We agree the experiment section was a little short, so we have made several updates to the format of the paper (including adding Appendices with related results) to allow for more details and discussion of the proposed analysis and experimentation.
>
> Detailed Comments:
> - Section 3.1: We have clarified the experimental setting, including how the objective functions are used. In Appendix A.2, we have provided details of exactly how the models were trained and the checkpoints were saved, including a link to source code.
> - For training DNNs, validation data is usually used to know when to stop training. Whereas, in this paper, we do not care about reaching a certain level of training accuracy, just that we have a large range of accuracies to benchmark. That is, if the network is overfit due to too much training, this will be reflected in the test accuracy (IoU), and serves as an important data point for us.
> - We have added Section 3.2 “Evaluation Procedure” and Appendix A.2.4 to clarify this.
>
> Questions:
> - We believe that since UNets are quite ubiquitous in medical imaging, metrics and initial demonstration on only UNets will still serve a broad audience. However, in Appendix A.1, we have added additional results on CNN classification models, including fully-convolutional models (albeit, not image-to-image networks). In general, we feel that the methods presented in this paper are particularly well-suited for spatial convolutional layers.  For cases with recurrent modules, multiple branches, or other sparse matrix operations, additional development may be required. We have updated the text to reflect this.
> - We have reformatted the paper to provide more details and discussion of experimentation, results, and trends, including a description of how the evaluation was performed (including sample size), comparison to existing PCA-based methods, and a new plot of metrics as a function of layer. The Appendix A.2.1 also includes details about how the experimental parameters were chosen.
> - In section 3.2, we have added comparisons to existing PCA-based clustering methods (mentioned in the Prior Work), which perform poorly. Additional results on CNN models are in Appendix A.1.

---

### Meta-Review · Area_Chair1 · 2021-03-27

**Recommendation:** Accept (Poster)

**Metareview:**

The authors address a hot topic, visualizing features and representation inside FCN to improve Interpretability, with an interesting methodology.

Even though the paper had initially some limitations, such as : (i) the experimental scope is limited (only one type of architecture tested), (ii) some technical aspects need to be discussed, (iii)  there are some clarity issues, the authors took the reviewer’s comment into account in their new paper version.

In addition the code is released to the public.

Thus I will follow the reviewer’s suggestion to recommend acceptance of this paper.

**Paper Type:**

methodological development

---

### Decision · Program_Chairs · 2021-03-31

**Decision:**

Accept

**Comment:**

Congratulations your paper has been selected as a long oral.